# Adaptive Sensor Selection for Power Efficient 3D Object Detection on Autonomous Driving

## Abstract

3D object detection for autonomous driving relies on multiple sensors, but their parallel use increases power consumption, limiting operational time and performance. In addition, not all scenarios require every sensor. An excessive number of sensors not only constrains inference performance but can also degrade accuracy by introducing noise. To achieve the Pareto optimality between accuracy and efficient performance, we introduce `AdaSensor`, an **Ada**ptive **Sensor** selection framework for power-efficient 3D object detection. The `AdaSensor` first designs the Mixture of Sensors (MoS) module that employs a lightweight sensor router to choose the necessary sensors during inference. This selective sensor activation reduces the computational load by processing fewer inputs and lowers power consumption by deactivating unused sensors. However, a naive MoS suffers from inference instability, rooted in the latency overhead from frequent sensor switching. To mitigate this, `AdaSensor` incorporates a novel non-congested switching policy to judiciously limit the switching frequency, which enhances system stability and efficiency while extending sensor lifetime. To demonstrate the effectiveness and efficiency of our method, we evaluate `AdaSensor` on the classical autonomous driving computing platform Nvidia Jetson Orin. On the nuScenes dataset, our method effectively reduces system power consumption by 11.7% and inference latency by 11.3%. The code will be made publicly available.

## 1 Introduction

3D object detection is crucial for autonomous driving. One important approach is the use of multi-modal and multi-sensor inputs (Chen et al., 2022b; Gao et al., 2023; Huang et al., 2021; Liu et al., 2023a). This is necessary due to the limitations of individual sensors. For example, cameras provide rich color and texture information, but struggle with depth estimation. LiDAR measures precise 3D geometry directly, yet provides less color information, and is less reliable in extreme weather, such as heavy rain or snow. Thus, fusing camera and LiDAR data is a popular method for 3D object detection in autonomous driving (Cui et al., 2021; Liu et al., 2023b).

Many efforts focus on using multi-sensor and multi-modal approaches for 3D object detection. Current popular methods in autonomous driving fall into two paradigms: *multi-camera fusion* and *LiDAR-camera fusion*. *Multi-camera fusion* combines outputs from multiple cameras to infer depth information and predict 3D positions (Xie et al., 2022; Liu et al., 2022b; Yang et al., 2023a). In contrast, *LiDAR-camera fusion* utilizes camera outputs along with depth information from LiDAR.

However, using more sensors introduces two critical challenges in real-world autonomous driving systems: ❶ **Power Consumption Upsurge**: Using more sensors increases computational demand, raising power consumption and inference latency. The computing platform and sensors can account for up to **77%** and **29%** of total power in an autonomous driving system (Katare et al., 2023). This power draw can reduce a vehicle's driving range by up to **12%** (Lin et al., 2018), posing a critical challenge for systems constrained by limited battery and computational capacity. ❷ **Over-Sensing**: According to different scenarios, not all sensors are necessary for perception. For example, on a two-lane freeway, at least exist one side of the car is not necessary. In addition, the human attention mechanism highlights the importance of focusing on critical information rather than processing all

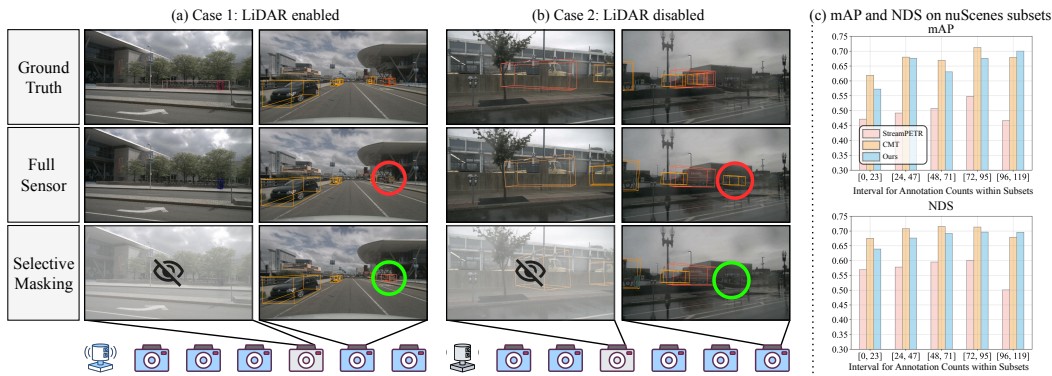

Figure 1: **Case study.** *(a), (b):* The Full Sensor detectors (*e.g.* CMT and StreamPETR) made incorrect predictions in both the left and right columns. After masking the left column camera, the right column's prediction became correct. This suggests the left column camera may have introduced noise to the model. *(c):* mAP and NDS scores on 5 nuScenes subsets divided by number of annotations. Higher number of annotations indicates a more complex scene. Our sensor selection outperforms full-sensor detector in the most complex subset, indicating its potential capacity.

available data. In the object detection task, oversatisfying information can dilute the attention on critical objects, decreasing accuracy (Wang et al., 2022). Therefore, we argue that the selective masking of sensor outputs provides a novel perspective to improving the 3D object detection Pareto optimality between power efficiency and accuracy.

Figure 1 (a) and (b) show that masking specific sensor outputs can improve 3D object detection accuracy. We divided the data into five subsets based on the number of objects. The construction and visualization of these subsets are detailed in Appendix E. A higher number of objects indicates a more complex scene. We tested the detectors in these different scenarios, both with and without sensor selection. Using sensor selection means we only use the outputs from the identified sensors for prediction. For without sensor selection, we use all available sensor outputs. As shown in Figure 1 (c), the mAP and NDS scores indicate sensor selection outperforms the full-sensor detector in the most complex scenes. These results emphasize the importance of sensor selection.

In light of the above challenges and observations, we introduce `AdaSensor`, an **Ada**ptive **Sensor** selection framework for power-efficient 3D object detection. Specifically, `AdaSensor` consists of two novel designs: ❶ *(Model Design)* `AdaSensor` introduces a lightweight Mixture of Sensors (MoS) module. It takes the front camera information to learning the adaptive sensor selection policy, which selects a suitable sensor subset. Our robustness detector then uses the output from the active sensors to perform 3D object detection. Therefore, we can directly eliminate the inactive sensors to save power consumption. This also reduces the amount of sensor information input to the network, which decreases the computational workload of our detector. However, the involve of the MoS is not benefit for deployment as the sensor switch will introduce additional latency. ❷ *(System Design)* Consequently, we introduce a novel inference strategy. It involve sensor-specific operational duration, which enables a non-congested sensor switch policy to avoid sensor switch latency and improve system execution stability. These two novel designs together improve the 3D object detection inference efficiency (less latency and better stability) while maintaining the accuracy. Our contributions can be summarized as follows:

- We propose an adaptive sensor selection framework for power-efficient 3D object detection. It uses a selected sensor subset for 3D object detection to save power consumption from both sensor running and detector computation.

- We present a Mixture of Sensors mechanism, which adaptively selects the adequate sensors' outputs for our detection model, thereby enabling a novel perspective for power efficiency 3D object detection on autonomous driving.

- We design a novel non-congested sensor switch policy to avoid sensor switching latency and improve inference latency stability.

- We evaluate `AdaSensor` on the Nvidia Jetson Orin, a real-world autonomous driving computing platform. The experimental results demonstrate that `AdaSensor` successfully improves 3D object detection efficiency performance and prediction accuracy. Specifically, `AdaSensor` achieves average reductions of 11.3% in latency, 23.7% in latency standard deviation, and 11.7% in system power consumption.

## 2 RELATED WORKS

**Camera-only 3D Object Detection** Detecting 3D objects from images is challenging due to the lack of direct depth information. Early monocular methods addressed this by inferring depth (Xu et al., 2021), using geometric priors (Lu et al., 2021), or designing specialized loss functions (Simonelli et al., 2020; Miao et al., 2021). The advent of multi-view datasets (Caesar et al., 2020; Sun et al., 2020) has spurred more advanced techniques, broadly classified into two categories. **Geometry-based** methods transform image features into Bird's-Eye View (BEV) representations. A seminal work, Lift-Splat-Shoot (LSS) (Philion & Fidler, 2020), achieves this by predicting a per-pixel depth distribution. Building upon LSS, BEVDet (Huang et al., 2021) adapted this transformation for detection, later enhanced by BEVDepth (Li et al., 2023) with explicit depth supervision and by BEVDet4D (Huang & Huang, 2022) with temporal alignment for improved velocity estimation. BEVHeight (Yang et al., 2023b) and BEVHeight++ (Yang et al., 2025) introduce predicting object height from the ground as a robust alternative to depth for view transformation, later fusing both height and depth cues to create a more comprehensive framework. **Transformer-based** methods leverage attention mechanisms to map perspective-view features to BEV. BEVFormer (Li et al., 2022) employs deformable attention to query features from multi-view images. Subsequent works like StreamPETR (Wang et al., 2023) and SparseBEV (Liu et al., 2023a) use sparse object queries and adaptive attention mechanisms, respectively, to efficiently model spatial and temporal context.

**LiDAR-Camera Fusion 3D Object Detection** Fusing the rich texture from cameras with precise depth from LiDAR can significantly improve detection accuracy and robustness. Fusion strategies are typically categorized as feature-level or proposal-level. **Feature-level** fusion aims to create a unified representation. Early works like PointPainting (Vora et al., 2020) decorated point clouds with semantic features from images. More recent approaches either transform both modalities into the BEV space before merging, as seen in BEVFusion (Liu et al., 2023b; Liang et al., 2022), or use point features to query and enhance corresponding image features (Gao et al., 2023; Chen et al., 2022a;b). Frameworks such as CMT (Yan et al., 2023) adapt transformers for multi-modal fusion, while others like SparseFusion (Li et al., 2024a) focus on improving efficiency. **Proposal-level** fusion first generates proposals from each modality before combining them. For instance, F-PointNet (Qi et al., 2018) uses 2D image detections to generate 3D frustums for narrowing down point cloud searches. Other methods (Li et al., 2024b; Xie et al., 2023) generate separate proposals from each sensor and then consolidate them. However, these approaches can sometimes favor one specific modality, potentially limiting the full benefits of fusion.

The proposed `AdaSensor` framework utilizes two distinct detector types. These are a Camera-only detector and a LiDAR-Camera Fusion detector. A trainable Mixture of Sensors (MoS) mechanism then selects the appropriate detector. This selection strategy aims to effectively balance detection accuracy with inference power consumption.

## 3 METHODS

The proposed `AdaSensor` includes two key components: the Mixture of Sensors (MoS) module (Section 3.1) and the non-congested sensor switching policy (Section 3.3). To work effectively with the MoS routing policy, we customized a robust training pipeline for the detector that can integrate with the MoS routing mechanism (Section 3.2).

Figure 2 illustrates the overall framework of `AdaSensor` in a data flow manner. Although we need to extract the front camera feature first, feature extraction will be performed for each input sensor. Thus, the additional computational cost lies only in the routers of our Mixture of Sensors module.

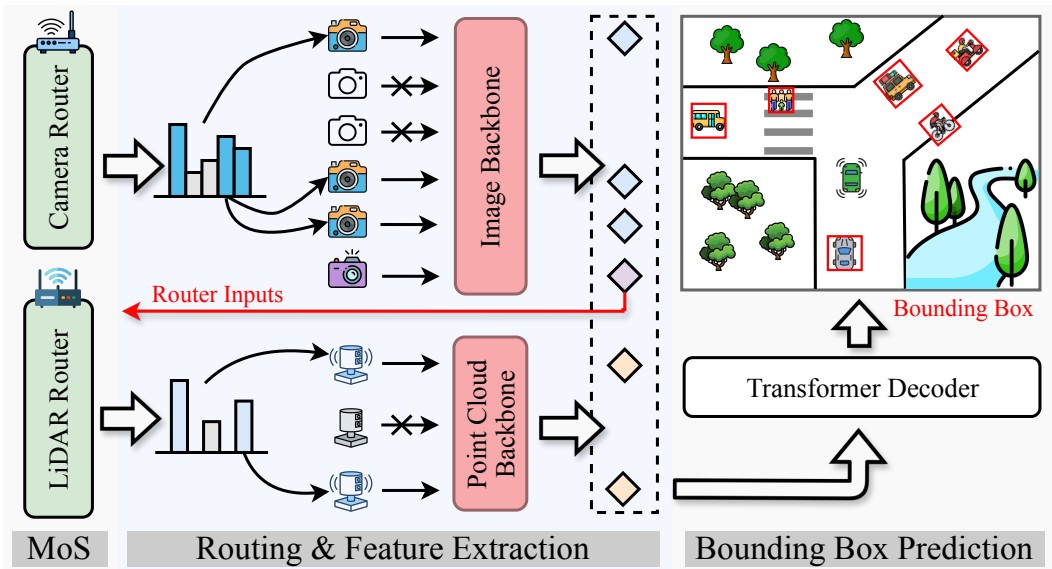

Figure 2: Overall Framework of `AdaSensor`. `AdaSensor` employs modality-specific routers. The Camera Router determines which camera will be used for detection. The LiDAR Router decides which LiDAR will be used (If only one LiDAR is available, the router will determine whether to use it). Both routers take input from the same front camera, but they process this information differently. `AdaSensor` first extracts front image features using the image backbone to make its sensor selections. Then, it uses the outputs from the selected sensors to extract sensor-specific features. Finally, these features are fed into the Transformer decoder for 3D bounding box prediction.

The extra computational cost from the two sensor routers is negligible. They are designed to be lightweight and depend solely on the front camera for input.

## 3.1 MIXTURE OF SENSORS

Considering the heterogeneity between LiDAR and camera, our Mixture of Sensors use two separate routers: **LiDAR Router** and **Camera Router**.

**LiDAR Router.** To make the LiDAR selection process learnable, we trained a LiDAR Router. Its function is similar to that of a Mixture of Experts (MoE) router in Large Language Models (LLMs) (Shazeer et al., 2017). Taking an image from the front camera as input, the LiDAR router routes to an arbitrary number of available LiDARs, or to a special "Zero LiDAR". Routing to "Zero LiDAR" means no LiDAR is used. Thus, in the single-LiDAR case, the router effectively decides whether to activate it. Notably, the LiDAR Router exclusively utilizes input from the front camera. This specific design consideration serves to minimize the sensor power consumption associated with this selection mechanism. The LiDAR Router is designed as a very lightwight 3-layer Vision Transformer (ViT) (Dosovitskiy et al., 2020) to further reduce the additional computational overhead.

To achieve this, we generate pseudo-labels for each training sample by comparing the detection mean Average Precision (mAP) of two state-of-the-art detectors: CMT (LiDAR-Camera) and Stream-PETR (Camera Only). We do not consider LiDAR-only detectors because the front camera will always be selected. Therefore, our minimum sensor selection subset consists of the front camera only, and there is no scenario where only LiDAR is selected. The pseudo-labeling process is as follows: For each sample, we first determine the maximum achievable accuracy for each LiDAR individually. This involves evaluating all possible LiDAR activation combinations and identifying the specific combination that yields its peak mAP on that sample. We then compare the peak accuracies achieved by CMT and StreamPETR. If StreamPETR's peak accuracy surpasses that of CMT, all LiDAR will be assigned "disabled". Finally, the LiDAR Router is trained with these pseudo-labels with a cross-entropy loss function. The detailed training setup is described at Section 4.2. This LiDAR Router identifies each LiDAR as an expert and it allows "Zero LiDAR" selection.

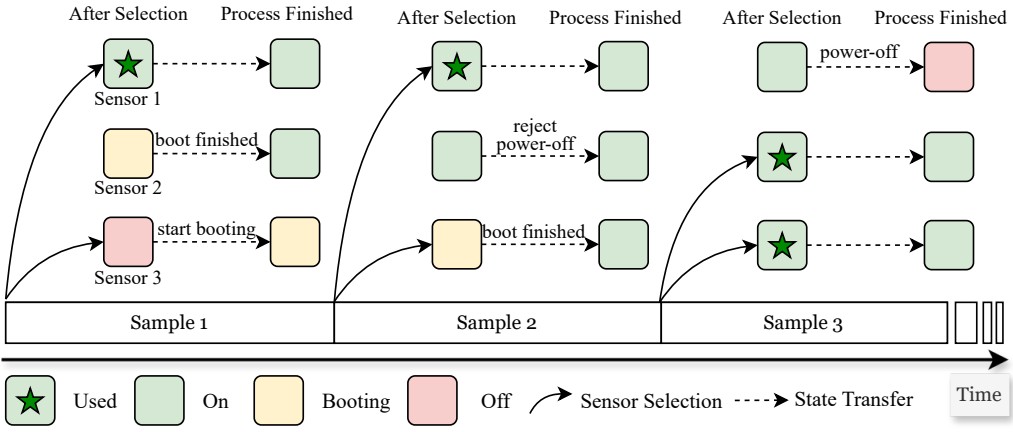

Figure 3: An example of our Dynamic Stability Policy with 3 sequential samples and 3 sensors. Sample 1 initially requires both Sensor 1 and Sensor 3. However, Sensor 3 is found to be inactive. Consequently, Sample 1 proceeds by utilizing only the operational Sensor 1, while Sensor 3 concurrently starts its booting process. Subsequently, Sample 2 does not select Sensor 2. Nevertheless, because Sensor 2 had been activated very recently, the system's policy rejects its deactivation. This ensures Sensor 2 remains powered on, allowing the subsequent Sample 3 to utilize it immediately without incurring a booting delay.

**Camera Router.** Similar to the LiDAR Router, the Camera Router predicts which cameras to activate based solely on input from the front camera. We present routing visualization in Appendix D. Note that the Camera Router is integrated as an additional embedded module within each detector. This embedded approach allows it to leverage the detector's inherent feature extraction capabilities. In practice, the Camera Router is implemented as a simple linear layer that processes front camera features extracted by the detector's image backbone to directly output selection probabilities for each camera. Such a lightweight structure results in almost negligible additional computational overhead.

Our Camera Router is conceptualized as a specialized MoE router, where each camera serves as an "expert". However, standard MoE training paradigms are ill-suited due to two challenges. First, unlike a typical MoE router that dispatches to a *fixed* number of experts, our router must dynamically select a *variable* number of cameras based on the input scene. Second, the features from selected cameras are forwarded directly to the downstream module, rather than being weighted by the router's continuous outputs. This direct-passing mechanism, while necessary for the downstream module, detaches the feature pathway from the routing. This severs the computational graph, preventing gradients from flowing to the router and making it untrainable with standard back-propagation.

To address these challenges, we introduce **Differentiable Top-$p$ Sampling**, an integrated routing mechanism that enables end-to-end training. This method first tackles the variable selection problem by leveraging the top-$p$ sampling principle (Huang et al., 2024). A linear routing network takes the front camera feature as input and generates a probability distribution $\mathcal{P}$ over all available cameras. Cameras are then iteratively selected in descending order of their probability until the cumulative sum exceeds a predefined threshold $p$ (set to $0.9$ in our experiments). This formulation allows the Camera Router to flexibly select a subset of cameras of variable size, thereby enhancing model flexibility without requiring ground truth for optimal sensor combinations.

To bridge the severed computational graph, we devise a custom gradient approximation inspired by the straight-through estimator (STE) (Bengio et al., 2013). In the forward pass, the continuous probabilities $\mathcal{P}$ are used to generate a binary mask $\mathbf{M}$, which assigns a value of 1 to selected cameras and 0 to others. The camera features are then multiplied by this mask, establishing a computational link without altering the feature values themselves. During the backward pass, the gradient is strategically channeled through this non-differentiable operation. Specifically, the gradient from a downstream loss $\mathcal{L}$ with respect to the binarized weights $\mathbf{W}$ is approximated by copying the gradient with respect to the initial probabilities $\mathcal{P}$ only for the selected cameras, *i.e.*, $\nabla_{\mathbf{W}}\mathcal{L} = (\nabla_{\mathcal{P}}\mathcal{L}) \odot \mathbf{M}$. This integrated approach effectively allows the Camera Router to be optimized jointly with the downstream detector. Further details are provided in Appendix B.3 and Algorithm 1.

As an embedded module within the detector, the Camera Router is trained directly using the original detection loss $L_{det}$ (Wang et al., 2023; Yan et al., 2023). Considering the expert collapse issue from the MoE training paradigm. We additionally incorporate the Load Balance Loss $L_b$ and Dynamic Loss $L_d$ (Huang et al., 2024) described in Appendix B.2. The overall loss is formulated as

$$L_{cam} = L_{det} + \alpha L_b + \beta L_d \tag{1}$$

For both CMT and StreamPETR, the loss weights $\alpha$ and $\beta$ are set by default to 0.1 and 0.01, respectively. Notably, the detector's parameters remain frozen during this Camera Router training phase, ensuring minimal training cost.

### 3.2 Power-efficient Robustness Training

Traditional methods for simulating sensor absence, like zeroing inputs or using special mask features (Chen et al., 2024; Yan et al., 2023), failed to reduce computational load as they didn't truly discard features from omitted sensors. Our approach combines CMT and StreamPETR models (which use a backbone for feature extraction followed by a Transformer (Vaswani et al., 2017) to fuse features from different sensors with cross attention), dynamically creates binary attention masks during training, preventing model queries from accessing features of inactive sensors. Consequently, during inference, feature extraction for these sensors is skipped, eliminating their computational load in both the backbone and Transformer. Detailed mechanism are provided in Appendix B.1.

### 3.3 Inference System Design

**Issue of Frequent Sensor Switching**  Frequent sensor activation and deactivation, driven by the sensor selection mechanism, can pose significant challenges. The time required for sensor booting introduces instability to the detector's inference latency. Such latency instability is unacceptable in autonomous driving systems (Liu et al., 2022a). Moreover, high switching frequencies can shorten the lifetime of the sensors. Consequently, from a system design perspective, it is necessary to constrain the sensor router's predictions to prevent the sensors from being switched too frequently.

**Baseline Policy**  Our baseline policy strictly follows the selections made by the sensor router. If a selected sensor is not yet active, the system idles until that sensor becomes operational. Only after the sensor is active does it use the corresponding backbone network to extract features. This operational strategy leads to the aforementioned issue of frequent sensor switching. Furthermore, it generates substantial GPU bubbles, thereby preventing the full utilization of computational resources.

**Dynamic Stability Policy**  To optimize the use of GPU computational resources and inference stability, our sensor activation policy is designed for efficiency. This policy incorporates two key strategies: First, we **prioritize active sensors to avoid delays.** If a selected sensor, such as a LiDAR, is not currently active, the system starts its booting process and concurrently employs a camera-only detector for the current task. For cameras, the system uses the intersection of the selected cameras and those already active. Second, we introduce a **Minimum Operational Duration (MOD)** for all sensors. In contrast to the baseline policy where unselected sensors are immediately deactivated, our system now actively rejects requests to shut down a sensor if it has been operational for less than this minimum duration. This effectively prevents frequent switching of sensors. We provide an example with 3 sequential samples and 3 sensors in Figure 3, illustrating various sensor state transfers.

## 4 Experiments

### 4.1 Experimental Setup

**Dataset**  We evaluate our `AdaSensor` on the nuScenes dataset (Caesar et al., 2020) for autonomous driving. It comprises 1000 driving scenes, each 20 seconds long. These scenes are divided into training, validation, and test sets, containing 700, 150, and 150 scenes, respectively. Data is collected from a sensor suite including 6 cameras, 1 LiDAR, and 5 radars, providing a full $360°$ field of view.

**Accuracy Metrics**  For 3D detection accuracy, we compare methods on the the nuScenes Detection Score (NDS), mean Average Precision (mAP), mean Average Translation Error (mATE), mean Average Scale Error (mASE), mean Average Orientation Error (mAOE), mean Average Velocity Error (mAVE) and mean Average Attribute Error (mAAE).

Table 1: Power consumption and boot time of sensors.

| Sensor Type | Model Info | Power (W) | Boot Time (s) |
|---|---|---|---|
| 32-line scanning LiDAR | RoboSense RS-Helios 1615 | 15.7 | 4.04 |
| Camera | Intel RealSense d455 | 1.2 | 3.21 |

**Performance Metrics** For inference system performance, we compare methods on the average latency and its standard deviation, where the latter indicates the stability of the system's inference. We also compared the average power consumption across different components: the sensors (denoted as $P_{\text{sen}}$), the computing platform (denoted as $P_{\text{plt}}$), and the system as a whole (denoted as $P_{\text{sys}}$).

## 4.2 IMPLEMENTATION DETAILS

**Training Setup of LiDAR Router** We use a 3-layer ViT with a patch size of 16, a feature dimension of 192, and 3 attention heads to build our LiDAR Router. Training is conducted based on the DeiT (Touvron et al., 2021) codebase. Input images are resized to a resolution of $224 \times 224$. We use a batch size of 64 and a learning rate of $1.0 \times 10^{-4}$, which is adjusted using a cosine decay schedule. Gradient clipping is set to a global norm threshold of 1.0. The model is trained using AdamW (Loshchilov & Hutter, 2017) optimizer for 50 epochs, with a warmup of 5 epochs. The entire training process takes only approximately 15 minutes on 4 RTX3090 GPUs.

**Training Setup of Camera Router** Before the training of the Camera Router, we first train the detectors employing the power-efficient robustness training technique described in Section 3.2. The difference between the original methods of CMT and StreamPETR is that CMT performed robustness training, while StreamPETR did not. For CMT, we adhere to its original training procedure, with the sole modification of replacing its robustness training approach with our proposed method. For StreamPETR, an initial training phase of 15 epochs is conducted strictly following its original method. Thereafter, our efficient robustness training technique is introduced for an additional 9-epoch training, resulting in total 24 epochs, which is aligned with the original StreamPETR. This training enhances the perception accuracy of both detectors when sensor features are missing.

Subsequently, all parameters of the detectors are fixed. The Camera Router module is then integrated following the image backbone of each detector. The Camera Router is subsequently trained for 5 epochs utilizing the loss function defined in Equation 1, while all other training configurations are kept consistent with those of the respective detectors.

**Inference System, Power and Latency Measurement** We measure the power consumption of required sensors for ten minutes and use the average value as the power estimation. Table 1 shows the model information, power consumption $P_{\{L,C\}}$, and boot time for the sensors employed in our study. For our baseline policy, we simulate sensor booting by making the inference program sleep for the duration of the boot time. Our inference system records the actual operating time $t$ for each sensor and uses the formula $\bar{P}_{\{L,C\}} = \frac{t}{T} P_{\{L,C\}}$ to estimate the average power consumption of each sensor. Here, $T$ denotes the total inference duration. To measure the power consumption of the computing platform (*e.g.*, an Nvidia Jetson Orin), we use the `tegrastats` tool to sample its real-time power consumption at a 10 Hz frequency. The average value is used as the platform's estimated power consumption. For latency, we measure the average wall-clock time per sample. The measured latency encompasses the entire data processing pipeline of a real-world autonomous driving system, including data pre-processing, model inference, and post-processing.

## 4.3 COMPARISON ON DETECTION BENCHMARKS

We performed end-to-end inference on the nuScenes validation set. Table 2 presents the accuracy results. Our `AdaSensor` achieves superior accuracy compared to camera-only detectors. When evaluated against LiDAR-Camera fusion detectors, `AdaSensor` demonstrates notable resource efficiency. It operates using an average of only **0.75 LiDARs** and **4.54 cameras**. Despite this reduced sensor usage, `AdaSensor` delivers 89.5% of the performance achieved by CMT (Yan et al., 2023). This result highlights an effective balance between detection accuracy and operational efficiency.

Table 2: 3D object detection accuracy on nuScenes **val** set. #L denotes average LiDAR count and #C denotes average camera count.

| Model | #L | #C | mAP↑ | NDS↑ | mATE↓ | mASE↓ | mAOE↓ | mAVE↓ | mAAE↓ |
|---|---|---|---|---|---|---|---|---|---|
| StreamPETR (Wang et al., 2023) | 0 | 6 | 0.501 | 0.584 | 0.572 | 0.261 | 0.390 | 0.250 | 0.199 |
| BEVNeXt (Li et al., 2024c) | 0 | 6 | 0.500 | 0.597 | 0.487 | 0.260 | 0.343 | 0.245 | 0.197 |
| CMT (Yan et al., 2023) | 1 | 6 | 0.679 | 0.708 | 0.315 | 0.252 | 0.314 | 0.252 | 0.178 |
| BEVFusion (Liu et al., 2023b) | 1 | 6 | 0.685 | 0.714 | - | - | - | - | - |
| DeepInteration (Yang et al., 2022) | 1 | 6 | 0.699 | 0.726 | - | - | - | - | - |
| AdaSensor | 0.75 | 4.54 | 0.608 | 0.667 | 0.365 | 0.252 | 0.310 | 0.260 | 0.189 |

Table 3: Average sensor counts on nuScenes **val** set. #L indicates average LiDAR count; #C_F, #C_FR, #C_FL, #C_B, #C_BL, #C_BR refer to average Front, Front-Right, Front-Left, Back, Back-Left, and Back-Right Camera counts respectively.

| Model | #L | #C_F | #C_FR | #C_FL | #C_B | #C_BL | #C_BR |
|---|---|---|---|---|---|---|---|
| CMT | 1 | 1 | 1 | 1 | 1 | 1 | 1 |
| StreamPETR | 0 | 1 | 1 | 1 | 1 | 1 | 1 |
| AdaSensor + BP | 0.83 | 1 | 0.84 | 0.68 | 0.75 | 0.84 | 0.74 |
| AdaSensor + DSP | 0.75 | 1 | 0.75 | 0.74 | 0.66 | 0.74 | 0.64 |

We further conducted experiments on the nuScenes-C benchmark (Dong et al., 2023) to demonstrate that `AdaSensor` maintains robustness and can even significantly improve accuracy in certain scenarios. The detailed results are provided in Appendix C.2.

## 4.4 PERFORMANCE EVALUATION

We conducted experiments on the nuScenes validation set to evaluate the performance of our `AdaSensor`, with both the Baseline Policy (BP) and our Dynamic Stability Policy (DSP). The average sensor counts are presented in Table 3 and the evaluation results are illustrated in Table 4.

In terms of speed and stability, our `AdaSensor` with DSP demonstrates a notable advantage. It achieves an **11.3%** reduction in latency and a **23.7%** decrease in latency standard deviation compared to the CMT baseline. This indicates a faster and more stable inference process. Conversely, while the `AdaSensor` with BP shows a substantial increase in latency (327%), its latency SD catastrophically rises by 1825%. This extreme instability suggests that the system frequently idles while waiting for sensor activation, leading to inefficient use of computational resources.

Regarding power consumption, both `AdaSensor` policies achieve considerable reductions. The `AdaSensor` with DSP reduces sensor power by **18.8%**, platform power by **7.1%**, and overall system power by **11.7%** relative to the CMT baseline, demonstrating the superiority of our adaptive sensor selection. Although the `AdaSensor` with BP achieves even lower power consumption figures (26.2% reduction in sensor power, 27.3% in platform power, and 26.9% in system power), this is largely a consequence of its aforementioned operational inefficiency. The significant idle periods, highlighted by the excessive latency standard deviation, mean that the GPU is underutilized, leading to lower but misleading power figures that do not reflect efficient system operation.

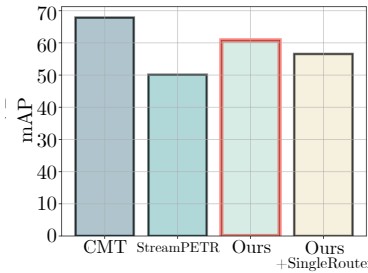

Figure 4: The ablation study of modality specific router *v.s.* single router (*e.g.* modality agnostic router) design. The red box denotes our proposed design.

## 4.5 ABLATIONS

**Is MoS really working?** To validate the effectiveness of the sensor selection strategy learned by MoS, we compare its performance (with MOD = 15.0) against two baselines: random selection and a heuristic approach. For the random selection baseline, the selection probability for each sensor is set to match the selection frequency (presented in Table 3) observed from MoS on the nuScenes validation set. The heuristic method determines camera activation for the current frame based on the bounding box predictions from the preceding frame; a camera is activated if its field of view

Table 4: 3D object detection performance metrics on nuScenes **val** set. BP denotes our Baseline Policy and DSP denotes our Dynamic Stability Policy. SD denotes Standard Deviation.

| Model | Latency↓ | Latency SD↓ | $P_{sen}$↓ | $P_{plt}$↓ | $P_{sys}$↓ |
|---|---|---|---|---|---|
| CMT | $0.397_{+00.0\%}$ | $0.0987_{+00.0\%}$ | $22.9_{+00.0\%}$ | $35.2_{+00.0\%}$ | $58.1_{+00.0\%}$ |
| StreamPETR | $0.430_{+08.3\%}$ | $0.0302_{-69.4\%}$ | $7.2_{-68.6\%}$ | $30.0_{-14.8\%}$ | $37.2_{-36.0\%}$ |
| AdaSensor + BP | $1.696_{+327\%}$ | $1.8995_{+1825\%}$ | $16.9_{-26.2\%}$ | $25.6_{-27.3\%}$ | $42.5_{-26.9\%}$ |
| AdaSensor + DSP | $0.352_{-11.3\%}$ | $0.0753_{-23.7\%}$ | $18.6_{-18.8\%}$ | $32.7_{-07.1\%}$ | $51.3_{-11.7\%}$ |

Table 5: 3D object detection accuracy and performance of MoS and baselines on nuScenes **val** set.

| Method | mAP↑ | NDS↑ | Latency↓ | Latency SD↓ | $P_{sen}$↓ | $P_{plt}$↓ | $P_{sys}$↓ |
|---|---|---|---|---|---|---|---|
| Random | 0.515 | 0.605 | 0.324 | 0.0883 | 17.1 | 32.8 | 49.9 |
| Heuristic | 0.596 | 0.663 | 0.378 | 0.2628 | 19.6 | 31.9 | 51.5 |
| MoS | 0.608 | 0.667 | 0.352 | 0.0753 | 18.6 | 32.7 | 51.3 |

Table 6: MOD values for our inference system. Our default setting is marked with  gray .

| MOD | mAP↑ | NDS↑ | $P_{sen}$↓ | $P_{plt}$↓ | $P_{sys}$↓ | Latency↓ | Latency SD↓ |
|---|---|---|---|---|---|---|---|
| 4.0 | 0.597 | 0.660 | 17.0 | 32.1 | 49.1 | 0.353 | 0.0854 |
| 10.0 | 0.605 | 0.664 | 18.0 | 32.6 | 50.6 | 0.356 | 0.0844 |
| 15.0 | 0.608 | 0.667 | 18.6 | 32.7 | 51.3 | 0.352 | 0.0753 |
| 20.0 | 0.613 | 0.669 | 19.1 | 32.7 | 51.8 | 0.359 | 0.0766 |

contained the center of at least one bounding box in the previous frame. The heuristic approach adopts the same LiDAR selections as MoS. In Table 5, MoS achieves the highest mAP and NDS, outperforming random selection by a large margin ($0.515$ $v.s.$ $0.608$ in mAP). Compared to the heuristic method, MoS achieves competitive accuracy while having lower and significantly more stable latency ($0.352$ $v.s.$ $0.378$ in latency, $0.0753$ $v.s.$ $0.2628$ in latency SD). Additional results for various MOD values are available in Appendix C.1, and a semi-formal analysis of the rules learned by MoS is provided in Appendix F. Collectively, these results confirm MoS learns an effective sensor selection strategy that enhances accuracy while reducing both power consumption and latency.

**What is an appropriate MOD?** To study the impact of the MOD value (detailed in Section 3.3) on `AdaSensor`, we experimented with different MODs and performed inference tests on the nuScenes validation set. As presented in Table 6, as the MOD value increases, the perception accuracy of `AdaSensor` gradually improves. Similarly, Sensor Power, Platform Power, and System Power also show a gradual increase. Taking into account the importance of latency and system stability, we choose MOD = 15.0, which yields the lowest latency and latency SD, as our default setting.

**Why use modal-specific routers?** To demonstrate the necessity of modal-specific routers, we conducted an experiment using a single modal-agnostic router. This router was trained with the pseudo-labels described in Section 3.1, using the modality combination that yields the highest mAP as the target. As shown in Figure 4, the modal-agnostic router achieves a significantly lower end-to-end mAP compared to our modal-specific routers ($56.6$ $v.s.$ $60.8$).

## 5 CONCLUSION

This paper introduces `AdaSensor`, a novel 3D detection framework utilizing adaptive sensor selection. The framework features a lightweight and trainable Mixture of Sensors (MoS) mechanism. This MoS mechanism intelligently chooses optimal sensors for 3D object detection. Notably, the selection process depends exclusively on input from the front camera. This design significantly lowers sensor power needs and overall computational energy use. We also developed a specialized inference system. This system works alongside `AdaSensor` to effectively mitigate inference latency. It also reduces system instability, a common issue arising from frequent sensor switching. Experimental evaluations on the Nvidia Jetson Orin confirm our approach. The results show that `AdaSensor` delivers substantial improvements in inference performance.

ETHICS STATEMENT

Our `AdaSensor` typically reduces energy consumption and extends the limited battery life of electric vehicles, directly translating to a lower carbon footprint. Lower power requirements can lead to longer operational ranges and potentially allow for the use of lower-cost hardware components (e.g., smaller batteries, less powerful processors). These factors can reduce the overall system cost of an autonomous vehicle, making the technology viable for a broader array of applications and economic contexts. However, adaptive systems choosing between saving power or activating an extra sensor in ambiguous, low-risk situations must have transparent trade-off principles. Otherwise, hidden biases in the decision logic can lead to unequal performance or risk.

REPRODUCIBILITY STATEMENT

The source code accompanying this paper is available in the supplementary ZIP file. All datasets used in this paper are publicly available. The experiments were conducted on an Nvidia Jetson Orin platform.

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

# A    THE USE OF LARGE LANGUAGE MODELS

Large Language Models were used solely for the purpose of text polishing and English grammar checking.

# B    ADDITIONAL IMPLEMENTATION DETAILS

## B.1    POWER-EFFICIENT ROBUSTNESS TRAINING

We simulate absent sensor features during training through the construction of attention masks. Specifically, models like CMT (Yan et al., 2023) and StreamPETR (Wang et al., 2023) employ a similar architecture: a backbone for feature extraction and a Transformer (Vaswani et al., 2017) subsequently fuses these extracted features with cross-attention. Specifically, the backbone features serve as keys and values, interacting with learnable object queries within the Transformer.

During training, backbone feature extraction occurs normally. However, binary attention masks are generated for randomly deactivated sensors. These masks zeros out the attention weights corresponding to deactivated sensors, preventing interaction between the Transformer and the corresponding keys and values from these deactivated sensors. Obviously, this masking strategy does not reduce computational load during the training phase. However, using attention masks in this manner is functionally equivalent to discarding features from the masked sensors, teaching the model to robustly handle the absence of their features in the cross-attention mechanism. Therefore, at inference time, significant computational savings are achieved. To be specific, we can completely bypass the feature extraction for deactivated sensors. We then create a shorter, dynamic key-value sequence for the transformer, using only features from the active sensors.

In a conclusion, the computation reduction includes the data pre-processing, backbone feature extraction, and associated Transformer computations for all deactivated sensors.

## B.2    LOADING BALANCING LOSS AND DYNAMIC LOSS

We adpot Load Balnacing Loss and Dynamic Loss from (Huang et al., 2024) to facilitate the training of our Mixture of Sensors (MoS).

**Load Balancing Loss**    The Load Balancing Loss, denoted as $L_b$, is introduced to prevent the router from disproportionately favoring a limited subset of sensors. This loss function is formulated as:

$$L_b = N * \sum_{i=1}^{N} f_i * Q_i \tag{2}$$

where $N$ is the total number of sensors, $f_i$ is the fraction of samples routed to sensor $s_i$ and $Q_i$ represents the average fraction of the router probability allocated for sensor $s_i$. For a batch containing $B$ samples, $f_i$ and $Q_i$ are calculated as:

$$f_i = \frac{1}{B} \sum_{j=1}^{B} 1\{s_i \in S^j\}, \quad Q_i = \frac{1}{B} \sum_{j=1}^{B} P_i^j \tag{3}$$

where $S^j$ is the sensors selected for sample $j$ and $P_i^j$ is the selecting probability of sensor $i$ on sample $j$. The indicator function $1\{s_i \in S^j\}$ is equal to 1 if sensor $s_i$ is among the selected sensors for sample $j$, and 0 otherwise.

**Dynamic Loss**    The Dynamic Loss, denoted as $L_d$, aims to prevent the router from assigning uniformly low confidence across all sensors for a given sample. Such an assignment would result in all sensors being selected, thereby failing to reduce energy consumption. This loss function as formulated as the entropy of the selecting probability $P = [P_1, P_2, \cdots, P_N]$:

$$L_d = - \sum_{i=1}^{N} P_i * \log P_i \tag{4}$$

Minimizing the entropy of the selection probability encourages the router to select as few necessary sensors as possible for a sample.

---

**Algorithm 1** Top-$p$ Sampling with Straight-Through Gradient

---

1: **Input:** Probabilities $\mathcal{P} \in \mathbb{R}^{B \times N}$ ($B$: batch size, $N$: number of sensors), threshold $p \in [0, 1]$.
2: **Output:** Binarized router weights with gradient $\mathbf{W} \in \{0, 1\}^{B \times N}$.

3: Initialize mask $\mathbf{M} \in \{0, 1\}^{B \times N}$ with all zeros.
4: **for** $i \leftarrow 1$ to $B$ **do**                                                    ▷ Iterate over each item in the batch
5:     $\mathbf{x} \leftarrow \mathcal{P}_{i,:}$                                                              ▷ Current row from $\mathcal{P}$
6:     $(\mathbf{s}, \mathbf{k}) \leftarrow$ SortDescending$(\mathbf{x})$          ▷ $\mathbf{s}$: sorted values, $\mathbf{k}$: original indices for elements in $\mathbf{s}$
7:     Initialize vector $\mathbf{c} \in \mathbb{R}^N$.                                             ▷ To store cumulative sums
8:     **for** $j \leftarrow 1$ to $N$ **do**                                                      ▷ Calculate cumulative sums
9:         $c_j \leftarrow \sum_{l=1}^{j} s_l$
10:     **end for**
11:     Create temporary sorted mask $\mathbf{m}' \in \{0, 1\}^N$.
12:     **for** $j \leftarrow 1$ to $N$ **do**
13:         **if** $c_j > p$ **then**
14:             $m'_j \leftarrow 1$
15:         **else**
16:             $m'_j \leftarrow 0$
17:         **end if**
18:     **end for**
19:     $\mathbf{M}_{i,:} \leftarrow$ RestoreOriginalOrder$(\mathbf{m}', \mathbf{k})$          ▷ Places elements of sorted mask $\mathbf{m}'$ into $\mathbf{M}_{i,:}$
    according to original indices $\mathbf{k}$
20: **end for**

21: $\mathbf{W} \leftarrow ((\mathcal{P} - \text{StopGradient}(\mathcal{P})) + \mathbf{1}^{B \times N}) \odot \mathbf{M}$
22: ▷ Forward: $\mathbf{W} = \mathbf{M}$. Backward: $\nabla_{\mathbf{W}}\mathcal{L} = (\nabla_{\mathcal{P}}\mathcal{L}) \odot \mathbf{M}$, where $\mathcal{L}$ denotes a downstream loss.
23: **return** $\mathbf{W}$

---

### B.3 DIFFERENTIABLE TOP-$p$ SAMPLING

To derive binarized weights from continuous probabilities while maintaining differentiability for end-to-end training, we employ the method detailed in Algorithm 1. This approach generates a binary mask $\mathbf{M}$ using a criterion related to top-$p$ selection (Huang et al., 2024).

In the forward pass, input probabilities $\mathcal{P}$ are first sorted in descending order, and their cumulative summations are calculated. A predefined threshold $p$ is then applied to these cumulative summations to determine the binary mask $\mathbf{M}$. The output binarized router weights $\mathbf{W}$ are set equivalent to this mask $\mathbf{M}$, i.e., $\mathbf{W} = \mathbf{M}$.

In the backward pass, we approximate the gradients similar to the straight-through estimator (Bengio et al., 2013). This allows the gradient of a downstream loss $\mathcal{L}$ with respect to the initial probabilities $\mathcal{P}$ to be effectively passed through the non-differentiable masking operation. Specifically, the gradient is computed as $\nabla_{\mathbf{W}}\mathcal{L} = (\nabla_{\mathcal{P}}\mathcal{L}) \odot \mathbf{M}$, ensuring that gradients are propagated only for elements where the corresponding mask value in $\mathbf{M}$ is non-zero. This mechanism enables the models incorporating this binarized weight generation to be trained end-to-end.

### B.4 TRAINING SETUP OF CAMERA ROUTER

For simplicity, we nonetheless directly utilized the original training setups of CMT and Stream-PETR. Note that Camera Router is the sole trainable module.

**CMT**  The Camera Router of CMT is trained with a global batch size of 16 on 4 RTX3090 GPUs. It is trained for total 5 epochs without CBGS using AdamW optimizer. The initial learning rate is $1.0 \times 10^{-4}$ and we follow the cycle learning rate policy. The weight decay is set to 0.01.

**StreamPETR**  The Camera Router of StreamPETR is trained with a global batch size of 16 on 4 RTX3090 GPUs. It is trained for total 5 epochs without CBGS using AdamW optimizer. The base learning rate is $4 \times 10^{-4}$ and we follow the cosine annealing learning rate policy with a 500-iteration warm-up. The weight decay is set to 0.01.

Table 7: 3D object detection accuracy and performance of MoS and baselines on nuScenes **val** set with MOD = 10.0.

| Method | mAP↑ | NDS↑ | Latency↓ | Latency SD↓ | $P_{sen}$↓ | $P_{plt}$↓ | $P_{sys}$↓ |
|--------|------|------|----------|-------------|------------|------------|------------|
| Random | 0.484 | 0.582 | 0.310 | 0.0906 | 15.8 | 32.6 | 48.4 |
| Heuristic | 0.592 | 0.661 | 0.378 | 0.2756 | 19.0 | 31.8 | 50.8 |
| MoS | 0.605 | 0.664 | 0.356 | 0.0844 | 18.0 | 32.6 | 50.6 |

Table 8: 3D object detection accuracy and performance of MoS and baselines on nuScenes **val** set with MOD = 20.0.

| Method | mAP↑ | NDS↑ | Latency↓ | Latency SD↓ | $P_{sen}$↓ | $P_{plt}$↓ | $P_{sys}$↓ |
|--------|------|------|----------|-------------|------------|------------|------------|
| Random | 0.538 | 0.618 | 0.335 | 0.0906 | 17.9 | 32.9 | 50.8 |
| Heuristic | 0.603 | 0.667 | 0.377 | 0.2417 | 20.1 | 32.0 | 52.1 |
| MoS | 0.613 | 0.669 | 0.359 | 0.0767 | 19.1 | 32.7 | 51.8 |

## C  ADDITIONAL EXPERIMENTS

### C.1  BASELINE EVALUATION WITH VARIOUS MOD

We evaluate the performance of MoS against two baselines, random selection and a heuristic approach, across three settings: MOD = {10.0, 15.0, 20.0}, with MOD = 15.0 as our default. The results are presented in Table 7, Table 5, and Table 8. Across all settings, the findings consistently support two conclusions:

(1) MoS significantly outperforms random selection in perception accuracy, even though both methods share the same underlying sensor selection probabilities. This indicates that MoS learns an effective, context-aware selection policy.

(2) Compared to the heuristic method, MoS achieves lower overall system power consumption ($P_{sys}$), primarily by reducing sensor power draw ($P_{sen}$) through more efficient sensor selections. Furthermore, MoS demonstrates lower and much more stable latency, avoiding the high instability of the heuristic approach, which is unacceptable for autonomous driving systems (Liu et al., 2022a).

In summary, these comparisons validate that MoS learns an effective sensor selection strategy that enhances perception accuracy while reducing both power consumption and latency.

### C.2  ROBUSTNESS EVALUATION WITH NUSCENES-C BENCHMARK

Table 9: NDS on nuScenes-C benchmark. The "Drop" column shows the performance degradation compared to the clean dataset.

| Corruption | CMT | CMT Drop (%) | AdaSensor+BP | AdaSensor+BP Drop (%) |
|------------|-----|--------------|--------------|----------------------|
| Clean | 70.8 | 0.0 | 68.0 | 0.0 |
| Snow | 67.7 | -4.4 | 66.9 | -1.6 |
| Rain | 69.9 | -1.3 | 62.9 | -7.5 |
| Fog | 69.2 | -2.3 | 55.3 | -18.7 |
| Sunlight | 49.3 | -30.4 | 61.3 | -9.9 |
| Gaussian | 66.1 | -6.6 | 64.7 | -4.9 |
| Uniform | 67.8 | -4.2 | 66.1 | -2.8 |
| Impulse | 66.5 | -6.1 | 65.0 | -4.4 |
| Moving Obj. | 67.3 | -4.9 | 63.3 | -6.9 |
| Obj. Shear | 68.2 | -3.7 | 64.1 | -5.7 |
| Obj. Scale | 67.8 | -4.2 | 63.3 | -6.9 |
| Obj. Rotation | 69.8 | -1.4 | 65.8 | -3.2 |
| Average | 66.3 | -6.4 | 63.5 | -6.6 |

Figure 5: Sensor co-activation heatmap on nuScenes **val** set. The CAM_FRONT is consistently active, thus its corresponding row indicates the activation probabilities of all sensors, highlighted by a red box.

To evaluate the generalization and robustness of AdaSensor, we test it on the nuScenes-C benchmark (Dong et al., 2023), which introduces corruptions like environmental variations, sensor noise, and object misalignment. We use the highest severity level for each corruption type to assess the NDS and compare the results with CMT. For this evaluation, we employ the AdaSensor+BP setting, as it allows for a direct assessment of how corruptions affect the router's decisions. Furthermore, since nuScenes-C corruptions are generated online at a slow pace, using the AdaSensor+DSP setting is technically impractical.

The results are presented in Table 9. The "Clean" row shows performance without any corruption, while the "Drop" column indicates the relative performance degradation under each corruption type. A smaller drop signifies greater robustness. On average, AdaSensor's performance degradation is comparable to that of CMT ($-6.6\%$ $v.s.$ $-6.4\%$). However, AdaSensor exhibits stronger robustness under certain conditions, achieving a smaller performance drop in Snow, Gaussian, Uniform, and Impulse corruptions. Notably, under the Sunlight corruption, AdaSensor's absolute NDS is significantly higher than CMT's, showcasing the ability of its dual-detector architecture to mitigate CMT's weaknesses in specific scenarios.

## D  Sensor Co-Activation Visualization

Figure 5 shows a sensor co-activation heatmap from the nuScenes validation set. This map covers different sensors including 1 LiDAR and 6 cameras. Notably, CAM_FRONT (the front camera) is the only sensor found to be consistently active. This is indicated by all values in its corresponding column being 1.000. Therefore, the values in the CAM_FRONT row represent the activation probability of each respective sensor. CAM_BACK and CAM_BACK_RIGHT have lower activation probabilities of 0.662 and 0.645 respectively, suggesting that information from these two viewpoints might be less critical for improving perception accuracy. The heatmap also displays a darker top-left region and a lighter bottom-right region. This pattern indicates that our MoS prefers to use the three forward-facing cameras. This preference likely exists because these particular views are more significant for autonomous driving tasks.

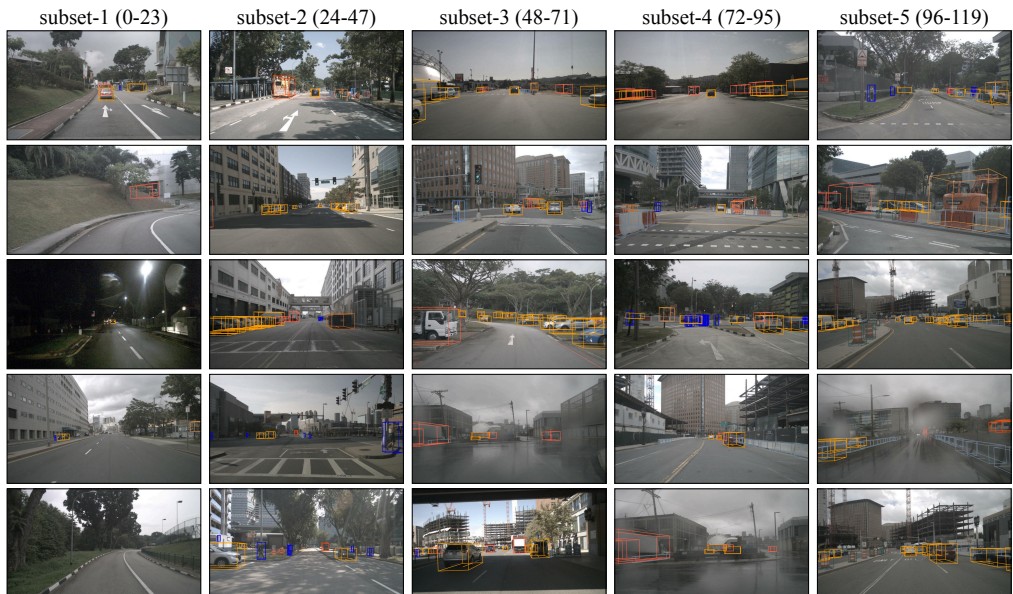

Figure 6: Visualization of `CAM_FRONT` images across different nuScenes subsets. Each column presents 5 sample images with ground truth annotations from a specific subset. The values in parentheses denote the interval of annotation counts within each subset.

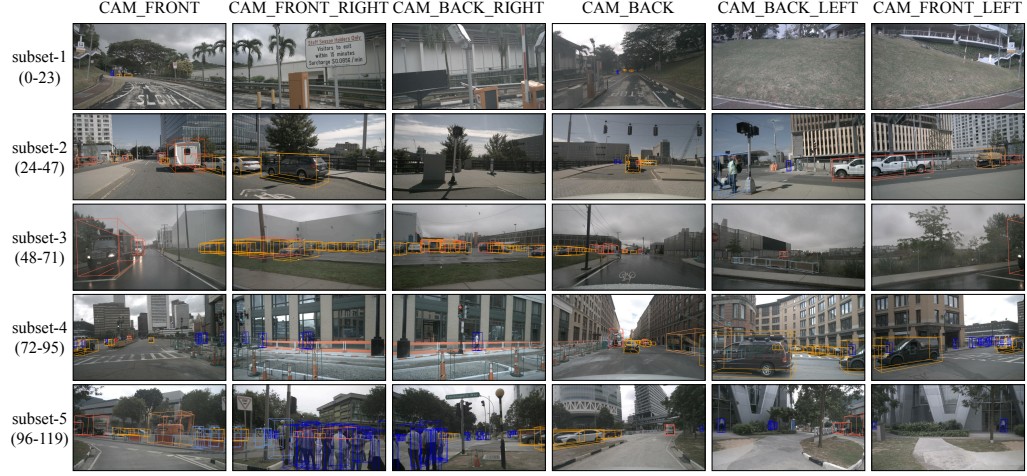

Figure 7: Visualization of all camera images across different nuScenes subsets. Each row contains images with ground truth annotations from the 6 cameras in one sample from the subset. The values in parentheses denote the interval of annotation counts within each subset.

# E NUSCENES SUBSET CONSTRUCTION AND VISUALIZATION

## E.1 SUBSET CONSTRUCTION

To facilitate a more granular analysis of model performance under varying conditions, we partitioned the nuScenes (Caesar et al., 2020) validation set based on scene complexity. We adopted the number of annotations present in each sample as a proxy for its complexity, where a higher annotation count generally indicates a more crowded and thus more challenging scene. This partitioning was

Table 10: Average sensor counts of MoS on nuScenes **val** set. #L indicates Average LiDAR count; #C_F, #C_FR, #C_FL, #C_B, #C_BL, #C_BR refer to Average Front, Front-Right, Front-Left, Back, Back-Left, and Back-Right Camera counts respectively.

| MOD | #L | #C_F | #C_FR | #C_FL | #C_B | #C_BL | #C_BR |
|---|---|---|---|---|---|---|---|
| 10.0 | 0.74 | 1 | 0.74 | 0.73 | 0.64 | 0.73 | 0.62 |
| 15.0 | 0.75 | 1 | 0.75 | 0.74 | 0.66 | 0.74 | 0.64 |
| 20.0 | 0.77 | 1 | 0.76 | 0.76 | 0.68 | 0.77 | 0.66 |

applied to the entire validation set. Initially, we determined the minimum and maximum number of annotations found across all these samples, considering all available object categories. This overall range of annotation counts was then divided into five equal intervals. Each sample from the validation set was subsequently assigned to one of these five subsets based on which interval its total annotation count fell into. These subsets correspond to annotation count intervals of $[0, 23]$, $[24, 47]$, $[48, 71]$, $[72, 95]$, and $[96, 119]$, respectively, ordered from the subset with the lowest annotation counts to the one with the highest. This stratification allows for evaluating model robustness across different levels of scene complexity.

### E.2 VISUALIZATION

Figure 6 shows `CAM_FRONT` images from five nuScenes subsets. Five samples are randomly chosen from each subset. This visualization clarifies that subset division depends solely on annotation count. Factors like time or weather are not considered for this division. The ground truth annotations shown allow for a qualitative assessment of annotation density. Note that the depicted annotations may not include all annotations for each sample.

Figure 7 shows images from all cameras within the five nuScenes subsets. One sample is randomly chosen from each subset. This visualization can effectively display all annotations associated with a single sample, clearly showing that annotation density varies significantly across camera views. For instance, in subset-1, `CAM_FRONT_RIGHT`, `CAM_BACK_RIGHT`, `CAM_BACK_LEFT`, and `CAM_FRONT_LEFT` have no annotations. Similarly, `CAM_FRONT_LEFT` in subset-5 shows very few annotations. Deactivating these cameras in such cases is a sensible approach to reduce power consumption.

## F LEARNED ROUTING RULES BY SENSOR ROUTERS

We perform a semi-formal analysis of the rules learned by the sensor routers using five subsets detailed in Appendix E. For this analysis, we collected the activation frequencies of the LiDAR and six cameras across these subsets and visualized them using heatmaps. Figure 8 presents the activation frequency of the LiDAR. The activation frequencies of the six cameras are shown for scenarios when the LiDAR is on (Figure 9) and off (Figure 10). Surprisingly, the analysis of these heatmaps reveals several intuitive rules:

1. **LiDAR usage strongly correlates with scene complexity.** As shown in Figure 8, the LiDAR activation frequency rises steadily from $0.704$ in the simplest scenes to $0.974$ in the most complex scenes. This confirms the intuition that the router learns to rely on LiDAR for robust perception in dense environments.

2. **The back camera is prioritized in complex scenes when LiDAR is active.** In Figure 9, the usage of `CAM_BACK` increases significantly to $0.853$ in the most complex bracket. This suggests a learned cooperative strategy where the system uses LiDAR for forward perception while activating the back camera to cover the rear.

3. **Camera selection is more targeted when LiDAR is disabled.** A comparison between Figure 9 and Figure 10 shows that camera activations are more uniform when LiDAR provides global context. In contrast, when LiDAR is off, the router must be more deliberate, leading to more specialized and varied camera selection patterns to compensate.

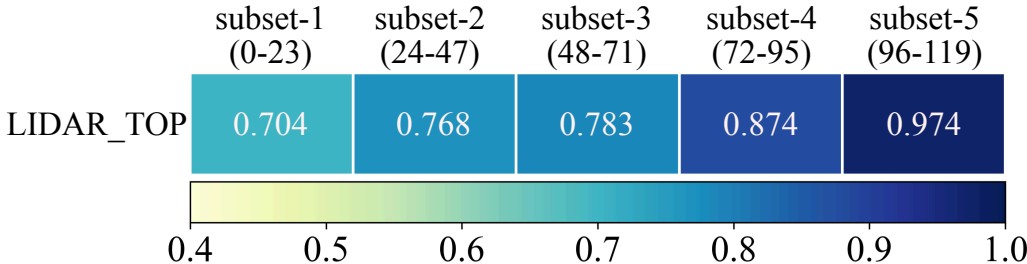

Figure 8: LiDAR activation heatmap on five nuScenes subsets. The results are produced by the LiDAR router.

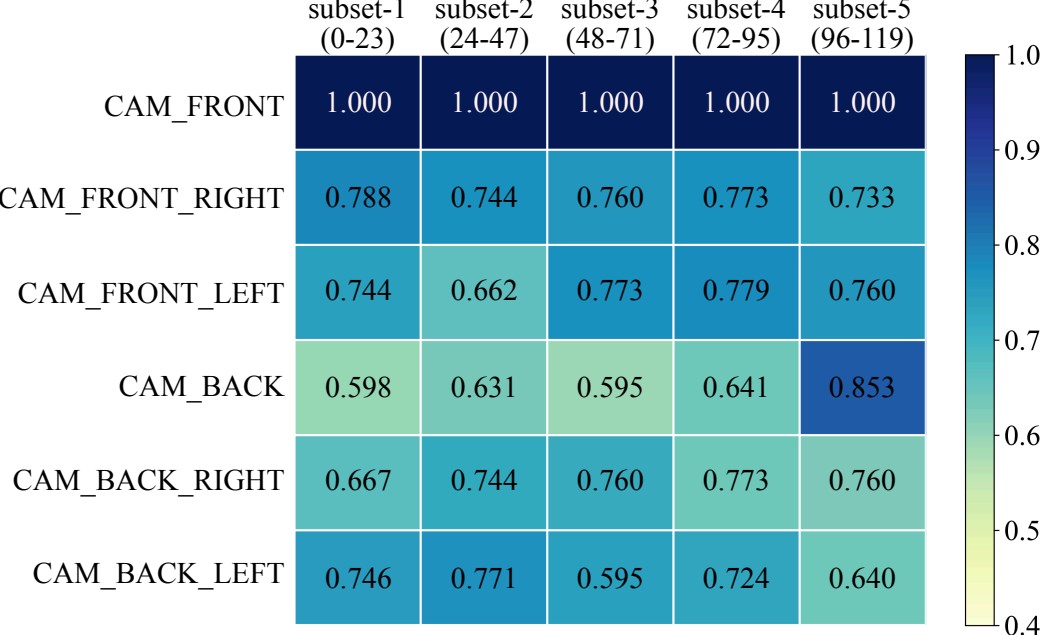

Figure 9: Camera activation heatmap on five nuScenes subsets (LiDAR enabled). The results are produced by the camera router of CMT.

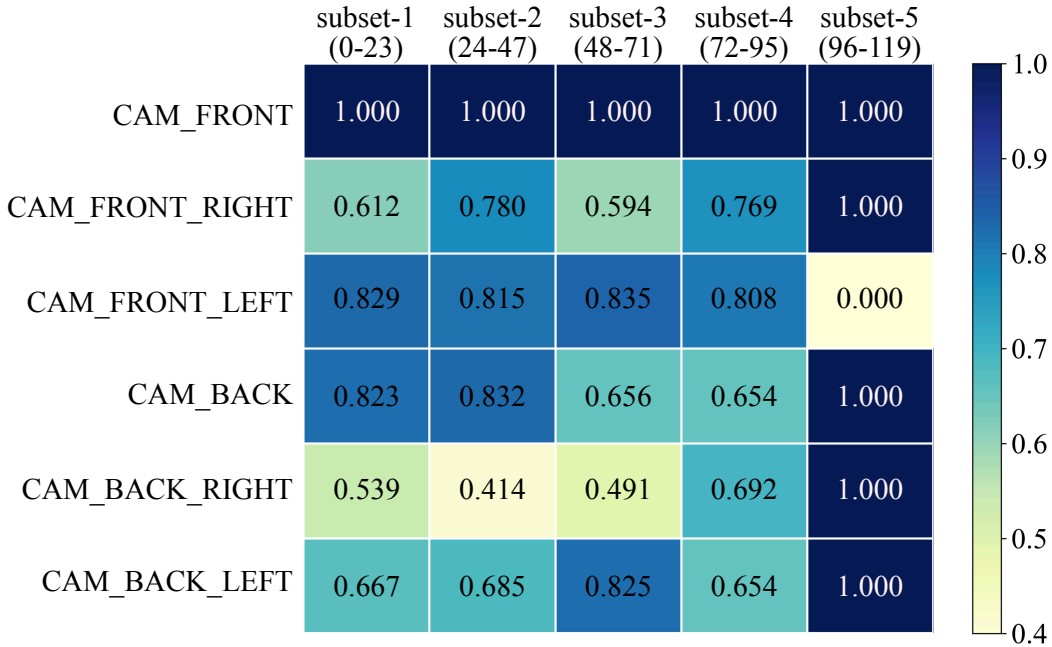

Figure 10: Camera activation heatmap on five nuScenes subsets (LiDAR disenabled). The results are produced by the camera router of StreamPETR.

