# OpenReview forum: "Adaptive Sensor Selection for Power Efficient 3D Object Detection on Autonomous Driving"
_ICLR.cc/2026/Conference — ICLR 2026 Conference Withdrawn Submission_

### Official Review · Reviewer_zRpA · 2025-10-18

**Soundness:** 3
**Presentation:** 3
**Contribution:** 2
**Rating:** 4
**Confidence:** 3

**Summary:**

The paper proposes AdaSensor, an adaptive sensor selection framework for 3D object detection in autonomous driving. It introduces a Mixture of Sensors (MoS) module that learns to activate only the most relevant sensors (cameras and LiDAR) using front-camera features, aiming to reduce power consumption and latency without significantly compromising accuracy. To address instability from frequent sensor switching, the authors propose a Dynamic Stability Policy (DSP) with a minimum operational duration constraint.

**Strengths:**

- Integrates perception model design (MoS) with real-time system policy (DSP), showing awareness of both algorithmic and deployment constraints.

- Power and latency measurements on Jetson Orin lend credibility to the “efficiency” claims beyond simulation.

- The co-activation heatmap (Fig. 5) and MOD ablation studies provide some transparency into the sensor selection behavior.

- Achieves nearly 90% of CMT’s accuracy using fewer sensors (0.75 LiDAR + 4.5 cams), confirming Pareto improvement in performance–efficiency space.

**Weaknesses:**

- Performance is limited. The proposed AdaSensor achieves only about 89.5% of the accuracy of the full-sensor baseline (CMT), while the overall inference speed improves by roughly 10%. This modest trade-off suggests that the framework mainly offers marginal efficiency gains at the cost of reduced perception performance, limiting its practical impact for high-accuracy autonomous perception systems.

- Both routers rely solely on front-camera input to determine global sensor activation, which risks poor decisions in non-frontal or occluded scenarios.

- LiDAR router is trained using pseudo-labels derived from other detectors’ performance (CMT vs. StreamPETR), introducing circular supervision and potential overfitting to benchmark distributions.

- Reported energy savings partly arise from deactivating sensors rather than improving computational efficiency—thus “efficiency” may conflate doing less work with doing work faster.

- On nuScenes-C, AdaSensor shows similar average degradation (−6.6% vs −6.4% for CMT), yet authors highlight selective improvements (e.g., sunlight) as general robustness.

**Questions:**

– Since DSP enforces minimum activation duration, how does this affect system responsiveness to sudden environmental change?

– Can the approach generalize to radar or thermal sensors, or is it limited to vision + LiDAR fusion?

– Could the router decisions be audited or explained at run time for regulatory compliance and debugging?

---

### Official Review · Reviewer_Y4Mu · 2025-10-29

**Soundness:** 3
**Presentation:** 3
**Contribution:** 2
**Rating:** 4
**Confidence:** 3

**Summary:**

This paper proposes an adaptive sensor selection framework to improve the energy efficiency of edge-based machine learning systems. The main idea is to dynamically activate only a subset of sensors depending on the context or task requirements, rather than using all sensors continuously.

**Strengths:**

1. The paper introduces a novel idea of adaptive sensor selection, which is relevant for energy-efficient autonomous driving systems.

2. The policy is small and suitable for edge devices with limited resources.

3. The experimental results demonstrate 11% energy savings while maintaining acceptable detection accuracy.

**Weaknesses:**

1. Only one small dataset is used

2. The paper does not show any tables to analyze the latency or the end-to-end response time.

3. For a large amount of sensors, the computational overhead of selection could be nontrivial.

**Questions:**

Overall, I think this paper address an important problem in the realm of autonomous driving, and the idea is interesting. I have a few comments and questions as follow.

How sensitive is the system to policy network errors? If the policy incorrectly deactivates an important sensor, how much does performance drop?

How is sensor activation latency handled? Does switching sensors on/off cause measurable delay?

How well does the method scale with more sensors? The computational overhead of the selection process could grow with the number of sensors, potentially offsetting energy savings.

Since some sensors are turned off, could this induce distribution shift or bias in downstream predictions?

---

### Official Review · Reviewer_9NMK · 2025-10-31

**Soundness:** 2
**Presentation:** 2
**Contribution:** 1
**Rating:** 2
**Confidence:** 4

**Summary:**

This paper observes that full sensor activation in autonomous driving is often redundant—many scenes can be accurately perceived with fewer sensors, reducing both computational and power costs. The proposed AdaSensor framework introduces a Mixture of Sensors (MoS) module that uses lightweight transformer-based routers—a 3-layer ViT for LiDAR and a differentiable Top-p sampling router for cameras—to dynamically select active sensors based on front-camera inputs. A power-efficient robustness training strategy masks inactive sensor features during training to maintain stability when sensors are disabled. To address latency instability from frequent switching, a Dynamic Stability Policy (DSP) imposes a minimum operational duration for each sensor to prevent boot-time delays.

**Strengths:**

This paper identifies the significant impact of the perception module on overall vehicle energy consumption, revealing the potential inefficiency of multi-sensor systems.

**Weaknesses:**

1. Safety risk of sensor deactivation: The paper’s core idea of turning off certain sensors to save energy is questionable in safety-critical autonomous driving scenarios. Real-world driving requires continuous 360° situational awareness— even human drivers rely on constant proximity sensing (e.g., ultrasonic anti-collision sensors). Disabling parts of the perception system means the vehicle is making decisions under incomplete environmental information, which could lead to unsafe behaviors or missed detections. The paper does not provide sufficient justification or safety analysis to demonstrate that such sensor reduction would not compromise driving safety.
2. Unclear and possibly marginal energy benefit: The paper’s discussion of sensor power consumption lacks clarity and proportional reasoning. If sensors account for only around 29% of total system power, and AdaSensor achieves roughly an 11% total reduction, the net gain may correspond to only about 3% absolute system-level savings. Considering the added complexity, potential latency overhead, and safety risks from frequent sensor switching, such a modest improvement may not justify the design trade-offs. The paper should provide a clearer energy breakdown and cost–benefit analysis to validate the claimed efficiency advantage.
3. Lack of consideration for sensor activation failure and system reliability: The paper does not discuss the risk of sensor startup or shutdown failures. In real vehicle systems, dynamically enabling or disabling sensors during driving is generally avoided because sensor booting can cause transient spikes in computational and power demand on the vehicle’s industrial computer. Such peaks may interfere with other modules or even lead to sensor initialization failures. Without addressing these potential reliability risks, the proposed dynamic activation strategy may be impractical for deployment in real autonomous driving systems.

**Questions:**

Given that autonomous driving is a safety-critical task requiring continuous 360° perception, how do you justify the practicality of intentionally disabling sensors during vehicle operation? Since this approach inherently reduces environmental awareness and introduces potential safety risks, what concrete evidence or safety mechanisms can you provide to ensure that selective sensor deactivation will not compromise driving reliability?

---

### Official Review · Reviewer_dsgC · 2025-11-01

**Soundness:** 3
**Presentation:** 2
**Contribution:** 3
**Rating:** 4
**Confidence:** 3

**Summary:**

This paper presents AdaSensor, an adaptive sensor selection framework designed to reduce power consumption and improve inference stability in multi-sensor perception systems for autonomous driving. The proposed method consists of two main components: the Mixture of Sensors (MoS) module, which dynamically determines whether to activate the LiDAR and which cameras to use based on front-camera features, trained through a differentiable Top-p sampling mechanism with two independent routers; and the Dynamic Stability Policy (DSP) module, which limits sensor switching frequency during inference by prioritizing already-active sensors and enforcing a minimum operational duration to reduce latency fluctuations. Experiments on the nuScenes dataset and NVIDIA Jetson Orin platform demonstrate that the approach reduces both power consumption and inference latency while maintaining detection accuracy, with additional ablation studies analyzing the effects of individual modules and parameter settings.

**Strengths:**

1. The paper addresses a practical and meaningful problem of energy efficiency and stability in multi-sensor systems for autonomous driving.
2. The experimental design is comprehensive, including main comparisons and ablation studies that support the main conclusions.
3. An anonymous code package is provided, which enhances the reproducibility and transparency of the work.

**Weaknesses:**

1. The overall novelty of the paper is limited. Both the MoS and DSP modules are mainly built upon existing methods (e.g., Mixture of Experts and Top-p Sampling) without enough innovation.
2. The paper states that sensor activation and deactivation are determined by the model’s output, yet the system is also designed to “prioritize active sensors.” This could lead to inconsistencies between the model’s prediction and the actual system behavior, and the underlying logic is not clearly explained.
3. The ablation study does not directly compare with a setting where all sensors remain active, making it difficult to quantify the precise trade-off between energy consumption and detection accuracy.

**Questions:**

1. The MoS module relies solely on front-camera features for sensor selection. Is it possible to use low-frequency or low-resolution LiDAR/radar signals as low-power triggering inputs?
2. In the DSP module, is there any mechanism to distinguish whether a currently active sensor tends to switch frequently (and should remain active) or is one that should stay inactive for longer periods? More explanations would be helpful.
3. Does the fixed Minimum Operational Duration (MOD) parameter in DSP need to be adapted for different driving scenarios? For instance, a large MOD may lead to unnecessary energy consumption on highways, while a small MOD could cause frequent switching in complex urban environments.

---

### Note · Authors · 2025-11-12

I have read and agree with the venue's withdrawal policy on behalf of myself and my co-authors.